# Sterol derivative binding to the orthosteric site causes conformational changes in an invertebrate Cys-loop receptor

Steven De Gieter[1,2], Casey I Gallagher[3], Eveline Wijckmans[3], Diletta Pasini[3], Chris Ulens[3]*[†], Rouslan G Efremov[1,2]*[†]

[1]Center for Structural Biology, Vlaams Instituut voor Biotechnologie, Brussels, Belgium; [2]Structural Biology Brussels, Department of Bioengineering Sciences, Vrije Universiteit Brussel, Brussels, Belgium; [3]Laboratory of Structural Neurobiology, Department of Cellular and Molecular Medicine, Katholieke Universiteit Leuven, Leuven, Belgium

**Abstract** Cys-loop receptors or pentameric ligand-gated ion channels are mediators of electro-chemical signaling throughout the animal kingdom. Because of their critical function in neurotransmission and high potential as drug targets, Cys-loop receptors from humans and closely related organisms have been thoroughly investigated, whereas molecular mechanisms of neurotransmission in invertebrates are less understood. When compared with vertebrates, the invertebrate genomes underwent a drastic expansion in the number of the nACh-like genes associated with receptors of unknown function. Understanding this diversity contributes to better insight into the evolution and possible functional divergence of these receptors. In this work, we studied orphan receptor Alpo4 from an extreme thermophile worm *Alvinella pompejana*. Sequence analysis points towards its remote relation to characterized nACh receptors. We solved the cryo-EM structure of the lopho-trochozoan nACh-like receptor in which a CHAPS molecule is tightly bound to the orthosteric site. We show that the binding of CHAPS leads to extending of the loop C at the orthosteric site and a quaternary twist between extracellular and transmembrane domains. Both the ligand binding site and the channel pore reveal unique features. These include a conserved Trp residue in loop B of the ligand binding site which is flipped into an apparent self-liganded state in the apo structure. The ion pore of Alpo4 is tightly constricted by a ring of methionines near the extracellular entryway of the channel pore. Our data provide a structural basis for a functional understanding of Alpo4 and hints towards new strategies for designing specific channel modulators.

*For correspondence:
chris.ulens@kuleuven.be (CU);
rouslan.efremov@vub.vib.be
(RGE)

[†]These authors contributed
equally to this work

**Competing interest:** The authors
declare that no competing
interests exist.

**Reviewing Editor:** Marcel P
Goldschen-Ohm, University of
Texas at Austin, United States

## Editor's evaluation

The authors solved cryoEM structural maps for the pLGIC homolog Alpo4 from an extreme thermophile worm in apo and CHAPS bound conditions. The data are convincing and valuable and reveal how a detergent can bind to the orthosteric site and induce a quaternary twist of the channel domains. A limitation is that it is difficult to relate these observations to channel function as the activating ligand for Alpo4 remains unknown.

## Introduction

Cys-loop receptors are allosterically regulated pentameric ligand-gated ion channels (pLGICs). Functionally pLGICs are classified into cation- and anion-selective receptors. The former class is exemplified by nicotinic acetylcholine (nAChRs) and serotonin (5-HT$_3$) receptors (*Corringer et al., 2012*).

Anion-selective receptors are comprised of glycine (GlyRs) and GABA$_A$ receptors. Because of their role in synaptic transmission, the inflammatory response and implication in diseases including gastrointestinal, psychiatric and cognitive disorders, startle disease, epilepsy, and smoking addiction (*Treiman, 2001*; *Albuquerque et al., 2009*; *Walstab et al., 2010*; *Ha and Richman, 2015*), human and mammalian homologs of pLGICs have been the subject of active research.

Recent advances in cryo-EM led to a rapid expansion of the structural data on nAChRs. The structures of three types of heteropentameric nicotinic receptors (α4β2, α3β4, and *Torpedo* muscle-nAChR) and one homopentameric receptor (α7) were determined (*Morales-Perez et al., 2016*; *Walsh et al., 2018*; *Gharpure et al., 2019*; *Rahman et al., 2022*; *Noviello et al., 2021*). For the α7 nicotinic receptor, structures of three major conformational states were solved: a resting, an agonist-bound activated, and an agonist-bound desensitized state. Cryo-EM structures also have been determined for 5-HT$_3$ receptors (*Basak et al., 2018*; *Polovinkin et al., 2018*; *Hassaine et al., 2014*, glycine receptors *Huang et al., 2015*; *Du et al., 2015*), and GABA$_A$ receptors (*Kim et al., 2020*; *Masiulis et al., 2019*; *Zhu et al., 2018*; *Laverty et al., 2019*; *Miller and Aricescu, 2014*). Currently, only one structure of a non-vertebrate ion channel, the glutamate-gated chloride channel (GluCl), from *C. elegans* has been solved. GluCl is the drug target for anthelmintics such as ivermectin highlighting the importance of structural characterization of non-vertebrate pLGICs (*Althoff et al., 2014*; *Hibbs and Gouaux, 2011*).

Lophotrochozoa comprises one of the largest groups in the animal kingdom and includes organisms such as annelids, mollusks, and platyhelminths (flatworms). Intriguingly, genome analysis revealed a massive expansion of nAChR genes in these organisms with 52 and 217 nAChR genes identified in mollusks and annelids, respectively (*Jiao et al., 2019*). This contrasts with the number of receptors found in organisms having an advanced nervous system, in which only 10–20 nAChRs are encoded in the genomes, for example, 17 in humans (*Walsh et al., 2018*). It has been speculated that the expansion in nAChR genes is a consequence of the adaptation to a stationary lifestyle in a dynamic environment (*Jiao et al., 2019*). The biological role of the additional nAChRs and their properties remain unknown. Characterization of these receptors may lead to discoveries of alternative neurotransmitters, new signaling pathways as well as a better understanding of the evolution of neurotransmission.

We have previously biochemically characterized seven invertebrate Cys-loop receptors, Alpo1-7, identified in the proteome of *Alvinella pompejana*, an annelid worm that inhabits the surroundings of hydrothermal vents and is the most extreme thermophilic invertebrate currently known (*Chevaldonné et al., 2000*; *Holder et al., 2013*). Among seven Alpo receptors, we identified two nAChR-like receptors (Alpo1 and Alpo4) and one Gly-like receptor (Alpo6), which were expressed and purified in amounts suitable for structural studies (*Wijckmans et al., 2016*). Alpo4 has 27–29% sequence identity with α-subunits of nAChRs and 25% with 5-HT$_3$ receptors. The high biochemical stability and preliminary characterization using negative stain electron microscopy suggested that Alpo4 was a promising target for structural studies (*Kocot et al., 2017*). However, despite exhaustive screening in different expression systems in combination with a compound library, including acetylcholine and serotonin, we could not identify the agonist/neurotransmitter for Alpo4, thereby limiting functional studies (*Wijckmans et al., 2016*). Because of its biochemical stability and unique position between nAChRs and 5-HT$_3$ receptors, we characterized the structure of Alpo4 using single-particle electron cryogenic microscopy (cryo-EM).

## Results

### Alpo4 is an isolated member of lophotrochozoan nAChRs

A massive expansion of nAChR genes in lophotrochozoans suggests their importance in functional diversity and adaptations. In *A. pompejana* the total number of nAChR genes is not known because its genome has not been fully sequenced. To get further insight into the relation of Alpo4 to other nAChRs we applied comparative genomic analysis. We performed a phylogenetic comparison of Alpo1-4 with nAChRs from annelids: *Capitella teleta* (CT), *Dimorphilus gyrociliatus* (DM), *Owenia fusiformis* (OW), *Hirudo verbana* (HV), *Helobdella robusta* (HR), and from mollusca: *Crassostrea virginca* (MV), *Crassostrea gigas* (CG), *Mizuhopecten yessoensis* (MY), *Pecten maximus* (PM) and *Pomacea canaliculata* (PC) (*Kocot et al., 2017*). A total of 649 sequences were grouped into 25 families (*Figure 1—figure supplement 1a*). In the phylogenetic tree some sequences cluster in molluscan-specific families (e.g.

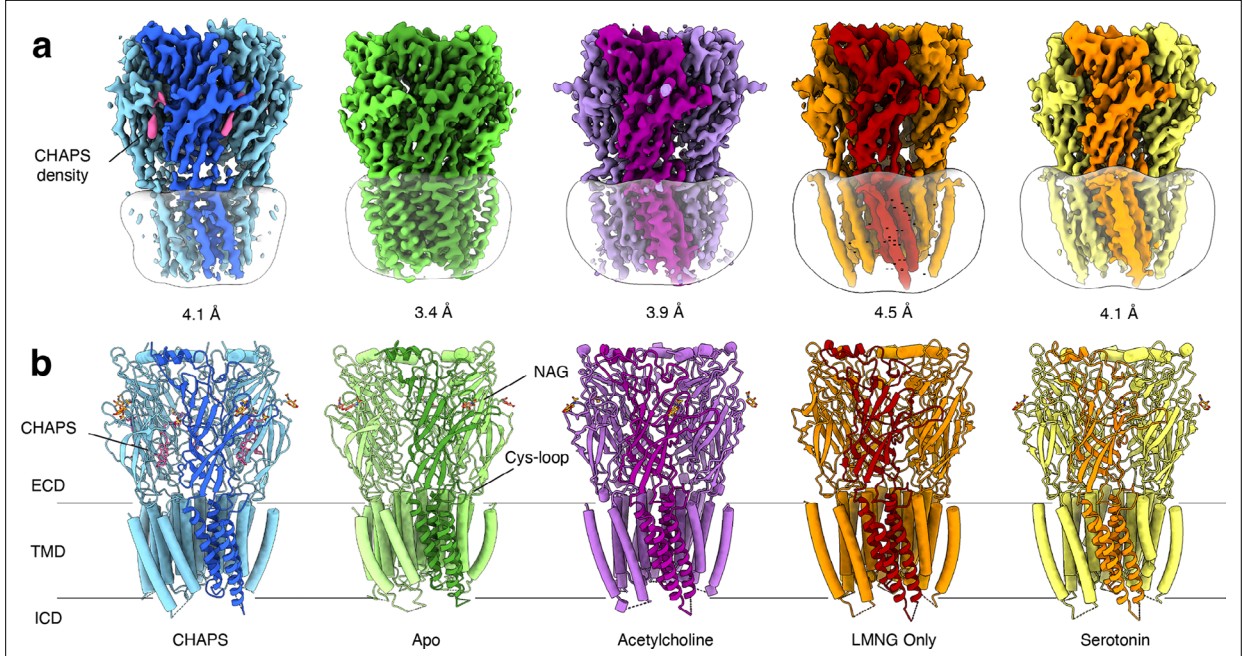

**Figure 1.** Overview of solved Alpo4 structures. (**a**) Electron cryogenic microscopy (Cryo-EM) reconstruction of apo Alpo4$^{CHAPS}$ (blue), Alpo4$^{APO}$ (green), Alpo4$^{ACH}$ (purple), Alpo4$^{APO\_LMNG}$ (orange), and Alpo4$^{SER}$ (yellow). The detergent micelle is shown in white surface representation. A monomer is shown in a darker shade. The density corresponding to bound CHAPS is shown in violet. (**b**) Side view of the atomic models shown in cartoon representation with NAG moieties shown as sticks. One subunit is highlighted. Bound CHAPS molecules are shown as sticks (violet).

The online version of this article includes the following figure supplement(s) for figure 1:

**Figure supplement 1.** Phylogenetic analysis of nicotinic acetylcholine (nAChR) in lophotrochozoans.

**Figure supplement 2.** Structure-based sequence alignment of cation-selective pentameric ligand-gated ion channels (pLGICs).

**Figure supplement 3.** Electron microscopy, classification, and 3D reconstruction for the Alpo4$^{CHAPS}$ dataset.

**Figure supplement 4.** Electron microscopy, classification, and 3D reconstruction for the Alpo4$^{APO}$ dataset.

**Figure supplement 5.** Quality of the electron cryogenic microscopy (cryo-EM) maps.

**Figure supplement 6.** Electron microscopy, classification, and 3D reconstruction for the Alpo4$^{APO\_LMNG}$ (LMNG only) dataset.

**Figure supplement 7.** Superposition of Alpo4 structures.

**Figure supplement 8.** Electron microscopy, classification, and 3D reconstruction for the Alpo4$^{ACH}$ (LMNG only) dataset.

**Figure supplement 9.** Electron microscopy, classification, and 3D reconstruction for the Alpo4$^{combined}$ (LMNG only) dataset.

**Figure supplement 10.** Electron microscopy, classification, and 3D reconstruction for the Alpo4$^{SER}$ (LMNG only) dataset.

58A, 58B, 61A, 62A, and 63A), whereas other families, like 33A, include members from all lophotrochozoan and have characteristic features of a vertebrate α7 subunit (*Li et al., 2016*).

Alpo1-4 were classified into different sequence families. Alpo2 and 3 are found in the families 32C and 41A (sequence identities of the closest homolog are 45% and 65%, respectively), both of which contain sequences from the genome of each included lophotrochozoan suggesting functional importance and conservation of these protein families within the clade (*Figure 1—figure supplement 1b*). Interestingly, in Polychaeta organisms, the characteristic vicinal disulfide in the tip of loop C required for ligand binding is not present in Alpo3-like nAChRs.

On the contrary, Alpo1 and Alpo4 (families 42A and 43A; sequence identities of the closest homolog 36% and 37%) do not clusters in the well-populated families (*Figure 1—figure supplement 1b*). This suggests either a unique function or a faster evolution. To further characterize Alpo4, we proceeded with its structural characterization by cryo-EM.

## Structure of Alpo4 reveals CHAPS bound to the orthosteric site

We purified Alpo4 in the detergent LMNG following the protocol established earlier (*Wijckmans et al., 2016*) and used cryo-EM to solve its structure. The map reconstructed to 4.1 Å resolution

confirmed that Alpo4 assembles into a homopentamer and has a conserved architecture of the pLGIC family (*Figure 1a*, *Table 1*). Each Alpo4 subunit is composed of a β-sandwich extracellular domain (ECD) and a transmembrane domain (TMD) made of four trans-membrane α-helices M1-M4 (*Figure 1b*, *Figure 1—figure supplement 2b*). Helices M2 contributed by each subunit are radially arranged around a central ion-conducting pore. The density for the intracellular domain (ICD), residues 308–412, was missing, and it was consequently not modeled (*Figure 1—figure supplement 2b*). An additional density next to the side chain of N167 on ECD was modeled as an N-acetylglucosamine (GlcNAc; *Figure 1b*, *Figure 1—figure supplement 2b*). Glutamine glycosylation at the structurally equivalent position was also found in the 5-HT$_3$ receptor, but not in nicotinic receptors (*Polovinkin et al., 2018*).

Although no ligand was added, an additional density was observed in the orthosteric binding site located at the interface between the ECDs (*Figure 1a*, *Figure 1—figure supplement 4a*). Despite the limited resolution, this density was well-resolved and consistent with a CHAPS molecule, a steroid-derived detergent. We further refer to this structure as Alpo4$^{CHAPS}$. CHAPS was present in the purified Alpo4 at a concentration of 0.007% (110 μM) because of its thermostabilizing effect on Alpo4 (*Wijckmans et al., 2016*). At 110 μM, the CHAPS concentration is 70-fold lower than the critical micellar concentration (CMC) of the detergent, suggesting a specific interaction with Alpo4 beyond modulating properties of the detergent belt and, therefore, is consistent with CHAPS binding to the orthosteric site (*Petroff et al., 2022*).

## Structure of the ECD and ligand-binding pocket with CHAPS

The ECD comprises an amino-terminal α-helix followed by 10 β-strands folded into a β-sandwich (*Figure 1b*, *Figure 1—figure supplement 2b*). CHAPS is bound at the ligand-binding pocket located at the interface of the principal (loops A-C) and the complementary subunit (loops D-F; *Figure 2a and d*). The CHAPS-Alpo4 interactions can be divided into two regions: the hydrophilic moiety and the sterol-binding moiety (*Figure 2—figure supplement 1a*). The hydrophilic moiety, in part formed by the dimethylammonio group, shares structural resemblance with carbachol and overlaps with the canonical Cys-loop receptor ligand-binding site, involving a group of highly conserved aromatic residues F103 (loop A), W159 (loop B), Y199, and Y205 (loop C) of the principal subunit and W65 (Loop D) of the complimentary subunit. Here, the quaternary ammonium group of the CHAPS molecule establishes a cation-π interaction with W159 (*Figure 2a and d*). This interaction is strikingly similar to the cation-π interactions observed with the quaternary ammonium group of the carbachol-bound AChBP structure (PDB: 1UV6), or the pyrrolidine nitrogen group in the nicotine-bound α4β2 nAChR structure (PDB: 5KXI) (*Figure 2b and e*). The interaction with the hydrophilic moiety is further stabilized by a salt bridge between the sulfonate group of CHAPS and Alpo4-specific Arg171 (*Figure 2a*).

The sterol ring moiety of CHAPS is composed of three cyclohexane and one cyclopentane ring and it fits into a hydrophobic crevice with a high shape-complementarity. The sterol ring interacts with residues F103 (loop A), N104, F137, V155, and W197 on the principal side and residues V48, K49, and D181 on the complementary side exclusively via Van der Waals contacts (*Figure 2a*, *Figure 2—figure supplement 1a*). This hydrophobic crevice is specific to Alpo4 and it is lined by poorly conserved residues. In other nAChRs, the pocket is narrower (*Figure 2—figure supplement 1b*) and is lined by multiple charged residues. These molecular interactions of CHAPS with Alpo4 explain why the binding of the detergent is specific.

## Structure of the ligand-binding pocket in the apo state

To gain further insight into the structural and ligand-binding properties of Alpo4, several structures were solved in the absence of CHAPS (*Figure 1*). This was accomplished in two ways. Either CHAPS was removed from the Alpo4$^{CHAPS}$ using size exclusion chromatography or we purified Alpo4 without any CHAPS present (*Table 1*). The highest resolution reconstruction of 3.4 Å was obtained from the protein sample that was depleted of CHAPS using size-exclusion chromatography (*Figure 1—figure supplement 4*). This reconstruction, which we refer to as the apo state (Alpo4$^{APO}$), allowed detailed modeling of the atomic structure of Alpo4 in most parts of the density (*Figure 1—figure supplement 5*). Despite the higher overall resolution of the reconstruction, the density corresponding to the tip of the C- and the F- loops was less well resolved which indicates their increased flexibility in the absence

**Table 1.** Statistics of cryo-EM data collection, data processing, and model refinement.

| Data deposition | | | | | | |
|---|---|---|---|---|---|---|
| Alpo4 ID: | Alpo4$^{CHAPS}$ | Alpo4$^{APO}$ | Alpo4$^{APO\_LMNG}$ | Alpo4$^{ACH}$ | Alpo4$^{COMB*}$ | Alpo4$^{SER}$ |
| PDB ID: | 8BYI | 8BXF | 8BX5 | 8BXB | 8BXE | 8BXD |
| EMDB ID: | EMD-16326 | EMD-16317 | EMD-16308 | EMD-16314 | EMD-16316 | EMD-16315 |
| **Data collection** | | | | | | |
| Microscope | JOEL CRYOARM300 | | | | | |
| Acceleration voltage [kV] | 300 | | | | | |
| Energy filter | In-column Omega energy filter | | | | | |
| Energy filter slit width [eV] | 20 | | | | | |
| Spherical aberration [mm] | 2.55 | | | | | |
| Magnification | 60 000 | | | | | |
| Detector | Gatan K2 | Gatan K3 | Gatan K3 | Gatan K3 | Gatan K3 | Gatan K3 |
| Refined pixel size [Å] | 0.782 | 0.784 | 0.7596 | 0.7596 | 0.7596 | 0.7596 |
| Exposure time [s] | | | 2.985 | 3.955 | 2.985 / 3.955 | 3.955 |
| Number of frames | 61 | 61 | 59 | 59 | 59 | 59 |
| Electron exposure [e⁻/Å2] | 37 | 30 | 45 | 59 | 45/59 | 59 |
| Defocus range [μm] | 1.6–2.8 | 0.8–2.8 | 1.0–2.4 | 1.0–2.4 | 1.0–2.4 | 1.0–2.4 |
| Collected images | 5003 | 7153 | 11 839 | 13 830 | 25 669 | 2200 |
| Used images | 2682 | 3754 | 9387 | 7816 | 13 647 | 1595 |
| Particles picked | 372 518 | 939 370 | 1 242 287 | 2 597 373 | 508 966 | 446 221 |
| | | | | | | |
| **Data processing** | | | | | | |
| Symmetry | C5 | C5 | C5 | C5 | C5 | C5 |
| Particles refined | 23 543 | 18 654 | 135 177 | 251 656 | 131 380 | 79 454 |
| Final resolution [Å], FSC = 0.143 | 4.1 | 3.4 | 4.2 | 3.9 | 3.9 | 6.2 |
| Sharpening B-factor [Å2] | –176 | –139 | –297 | –170 | –235 | –1062 |
| Local resolution range [Å] | 3.7–4.6 | 2.9–5.0 | 3.5–7.7 | 3.2–5.8 | 3.2–5.8 | 5.5–7.0 |
| | | | | | | |
| **Model refinement** | | | | | | |
| Refinement package | PHENIX 1.19 | | | | | |
| Initial model used | 6HIQ | Alpo4$^{CHAPS}$ | Alpo4$^{ACH}$ | Alpo4$^{APO}$ | Alpo4$^{ACH}$ | Alpo4$^{ACH}$ |
| Model resolution [Å], FSC = 0.5 | 4.2 | 3.9 | 4.5 | 4.2 | 4.1 | |
| Model composition | | | | | | |

*Table 1 continued on next page*

*Table 1 continued*

| Data deposition | | | | | |
|---|---|---|---|---|---|
| Non-hydrogen protein atoms | 13 321 | 12 692 | 12 550 | 12 547 | 12 553 |
| Protein residues | 1600 | 1630 | 1630 | 1630 | 1600 |
| Ligands | 131 | 31 | 31 | 31 | 31 |
| **B-factors mean [Å2]** | | | | | |
| Protein | 63 | 93 | 203 | 54 | 93 |
| Ligand | 40 | 108 | 199 | 54 | 111 |
| **R.M.S deviations** | | | | | |
| Bond lengths (Å) | 0.002 | 0.005 | 0.003 | 0.004 | 0.004 |
| Bond angles (°) | 0.688 | 1.240 | 0.736 | 0.779 | 1.055 |
| **Validation** | | | | | |
| Molprobity score | 1.9 | 1.6 | 1.8 | 2.3 | 1.9 |
| Clashscore | 24.1 | 4.8 | 14.1 | 7.2 | 12.5 |
| Poor rotamers (%) | 0 | 0.4 | 0 | 5.5 | 0 |
| **Ramachandran plot** | | | | | |
| Favored (%) | 98 | 96 | 97 | 96 | 95 |
| Allowed (%) | 2 | 4 | 3 | 4 | 5 |
| Disallowed (%) | 0 | 0 | 0 | 0 | 0 |

*Combined Alpo4$^{ACH}$ and Alpo4$^{APO}$.

of a ligand. At low sigma levels, residual densities were observed in the ligand-binding and the sterol-binding pockets suggesting that residual CHAPS was bound at low occupancy.

Similar results were obtained from the sample prepared in LMNG without CHAPS, reconstructed to 4.1 Å (*Figure 1—figure supplement 6*). This structure is essentially identical to the apo state with (RMSD of 0.9 Å) (*Figure 1—figure supplement 7*). No residual density was observed in the ligand-binding pocket in the later reconstruction, supporting the assignment of the 3.4 Å map as an apo state. The conformation of the aromatic residues constituting the ligand-binding pocket differed from the CHAPS-bound state. Specifically, W159 (loop B) flips and forms a cation-π interaction with H131, while being pinched by W65 (loop D) (*Figure 2f*). In this conformation, W159 can no longer form a cation-π interaction with an external ligand, therefore, it is tempting to speculate that this conformation represents a 'self-liganded' state.

Given that Alpo4 shares a structural resemblance to the α4β2 nAChR and the 5-HT$_3$R, we investigated whether acetylcholine or serotonin binds into the ligand-binding site. To this end, structures of Alpo4 purified in the absence of CHAPS and with an added, 1 mM acetylcholine or 1 mM serotonin were solved to a resolution of 3.9 Å and 6.2 Å, respectively (*Figure 1*, *Figure 1—figure supplements 8 and 10*). Their overall conformation was identical to that of the apo state with an overall RMSD of 0.87 and 0.88 Å, respectively. No density in the ligand-binding site was observed in the reconstructions. Although 6.2 Å resolution is too low to interpret the density of serotonin, the overall quaternary structure was identical to that of the apo state. The absence of a quaternary twist expected for a desensitized conformation suggests that serotonin was not bound.

The cryo-EM reconstructions of Alpo4 obtained in the presence of acetylcholine and serotonin suggest that neither of the neurotransmitters binds Alpo4. This agrees with our electrophysiological experiments in various expression systems (*Xenopus* oocytes, HEK cells, lipid vesicles) indicating no agonist response to acetylcholine or serotonin (data not shown).

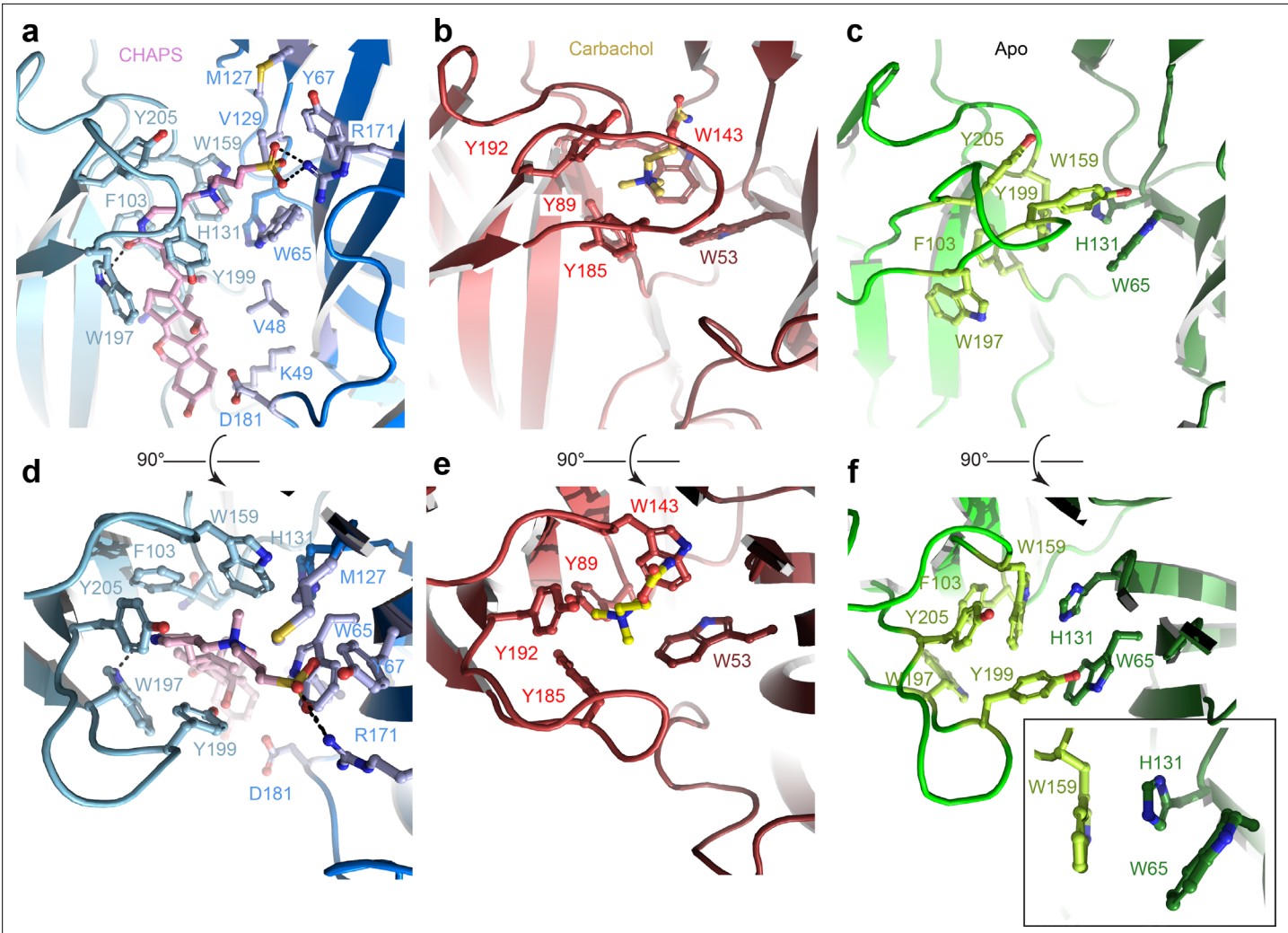

**Figure 2.** Orthosteric binding site of Alpo4. (**a, d**) Ligand binding pocket of Alpo4 with bound CHAPS. The residues interacting with CHAPS are shown as sticks. (**b, e**) Similar views of the ligand binding site in the acetylcholine binding protein (AChBP) in complex with carbachol (PDB code 1UV6). (**c, f**) Ligand binding pocket of Alpo4 in apo state. Residues constituting the aromatic cage are shown as sticks. The self-liganded state of W159 is shown as an inset in sticks representation. The principal subunit is colored in a lighter shade and complimentary in a darker.

The online version of this article includes the following figure supplement(s) for figure 2:

**Figure supplement 1.** Sterol-binding pocket in Alpo4 and other pentameric ligand-gated ion channels (pLGICs).

**Figure supplement 2.** Best docked poses of 12 compounds in Alpo4.

## Conformational changes upon binding of CHAPS

A comparison of the CHAPS-bound structure with the apo state reveals concerted conformational changes. In addition to the local rearrangements of side chains in the ligand-binding site (described above), we observe a clear change in the quaternary conformation of Alpo4 (*Figure 3*, *Video 1*). Upon binding of CHAPS, the ECD rotates 9° clockwise relative to the TMD (when viewed from the extracellular side; *Figure 3a and b*). The binding of CHAPS is associated with local conformational changes in the ECD. First, the tip of the loop C (residues 197–205) shifts by about 3 Å and extends (*Figure 3a and d*) even though its density in the apo state is somewhat ambiguous (*Figure 1—figure supplement 5*). The loop movement is accompanied by changes in the orientation of aromatic sidechains (Y195, W197, and Y199) that allow accommodating the zwitterionic moiety of the CHAPS molecule (*Figures 2 and 3d*; *Video 1*). On the complimentary subunit, loop F (residues 171–184) shifts by 3–5 Å toward the sterol group of CHAPS. This results in a small (~1 Å) rearrangement of the ECD-TMD linker (*Figure 3a and e*). The ECD protomers show concerted movements as rigid bodies. They rotate

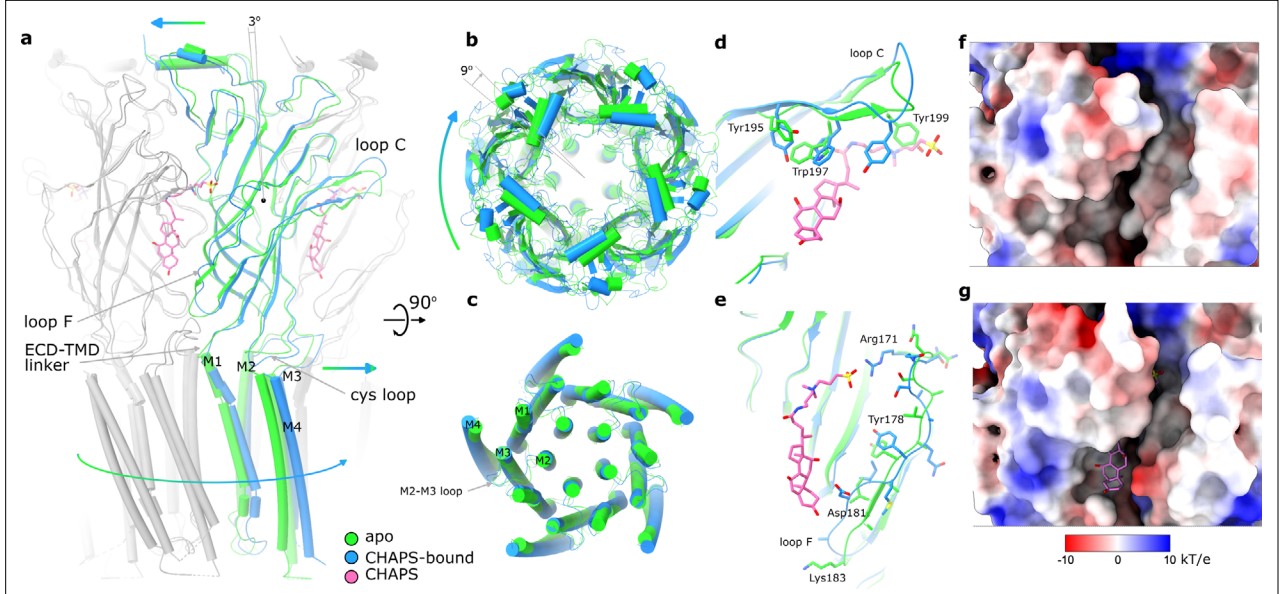

**Figure 3.** Conformational changes between apo and CHAPS-bound Alpo4. (**a**) Structures of apo (green) and CHAPS-bound (light blue) Alpo4 are overlayed. The extracellular domain (ECD) was aligned between the structures. Bound CHAPS is shown in pink for reference. Only one subunit from each pentamer is colored, others are shown in gray for clarity. The binding of CHAPS results in an around 3 degrees clockwise rotation. The approximate rotation center is indicated by the black dot. The rotation of the membrane domain is indicated by an arrow. (**b**) Relative rotation of transmembrane domain (TMD) and ECD. The structures are aligned to TMDs. (**c**) Same as panel (**b**) but only TMDs are shown. Changes in TMD associated with CHAPS binding are minor. (**d, e**) Close-up of conformational changes in loops C and F, respectively. (**f, g**) Surface electrostatics is shown around sterol-binding grooves for apo (**f**) and CHAPS-bound states (**g**).

The online version of this article includes the following figure supplement(s) for figure 3:

**Figure supplement 1.** Sterol binding groove is a ligand-binding site suitable for allosteric modulation.

by ~3 degrees around the domain center such that the apical regions of ECD move in the direction of neighboring ECD subunit in a clockwise fashion, whereases TMD-facing ends move counterclockwise (*Figure 3a*, *Video 1*). The whole ECD assembly rearranges as tightly packed domino tiles. The quaternary rearrangements preserve the hydrophobic CHAPS-binding groove (*Figure 3f and g*) even though its width changes.

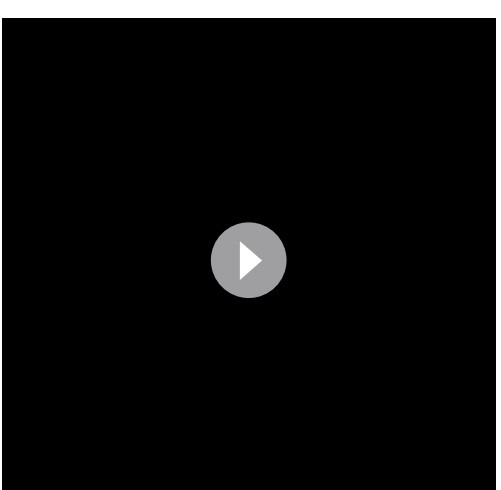

**Video 1.** Transition of Alpo4^APO to Alpo4^CHAPS viewed from the extracellular side, side view, the C-loop, and the F-loop.

https://elifesciences.org/articles/86029/figures#video1

We can speculate that the combination of CHAPS-induced rigid body tilt of the individual ECDs relative to each other and local rearrangements in the F loop leads to the rotation of TMD relative to ECD. This rotation is accommodated by the bending of the first two helical turns of the M1 helix, between 1 and 3 Å, an extension of the FPF motif on the Cys-loop, and a 3–5 Å shift of M2-M3 loop that follows rigid body movement rotation of the Cys-loop (*Figure 3a and c*; *Video 1*).

A quaternary twist is associated with gating transitions in characterized pLGICs (*Noviello et al., 2021*; *Basak et al., 2018*; *Polovinkin et al., 2018*; *Zarkadas et al., 2022*; *Petroff et al., 2022*; *Sauguet et al., 2014*; *Kumar et al., 2020*). Quaternary changes in Alpo4 induced upon CHAPS binding and those associated with the activation of related α7 nACh and 5-HT₃ receptors induced rotation of ECD relative to TMD in the same direction, however, the shifts of principal relative to complementary subunits were

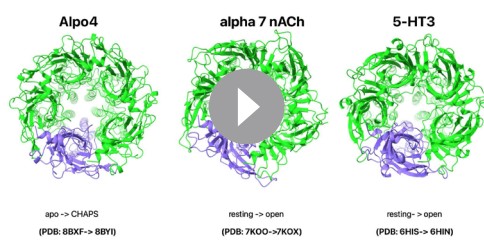

Alpo4 alpha 7 nACh 5-HT3

apo -> CHAPS resting -> open resting- > open
(PDB: 8BXF-> 8BYI) (PDB: 7KOO->7KOX) (PDB: 6HIS-> 6HIN)

**Video 2.** Quaternary conformational changes in Alpo4 upon binding of CHAPS are shown along with quaternary changes in α7 nicotinic acetylcholine (nACh) and 5-HT$_3$ receptors upon transition from resting to active state. The channels were aligned to the extracellular surface of the pore to show rotation of extracellular domain (ECD) relative to transmembrane domain (TMD) and to ECD of one subunit to show relative movements of ECDs. One subunit is shown in purple.

https://elifesciences.org/articles/86029/figures#video2

different (*Video 2*). In Alpo4, the complementary subunit slides upward whereas in the two other channels, it consistently shifts towards the principal subunit and tilts relative to the TMD. The tilt is less pronounced in Alpo4 which is probably why it does not lead to the pore dilation.

## Structure of the pore domain

In the TMD, all four TM helices are well resolved allowing for unambiguous assignment of the helix register. The densities for the M1-M3 helices are of excellent quality whereas the peripheral M4 displayed higher mobility (*Figure 1—figure supplement 5*). The ion pore is located along the fivefold rotational symmetry axis and is formed exclusively by the M2 helices. It has a circa 15 Å long hydrophobic patch in the outer leaflet of the membrane formed by three helical turns (residues 9'L, 13'L, and 16'M; *Figure 4a*). On the extracellular side, the hydrophobic region is flanked by negatively charged aspartate residues, 20'D, whereas on the intracellular side glycines G'6 create a cavity within the pore (*Figure 4b*) followed by rings of threonines (2'T) and conserved glutamates (–1'E) which usually play a role of the selectivity filter in cation-selective pLGICs (*Yonekura et al., 2015*). Thus, the charge distribution along the pore is consistent with Alpo4 being selective for cations.

A highly unusual feature in Alpo4 is the presence of bulky M265 residues at the 16' position. It forms the narrowest and most hydrophobic constriction (diameter of 2.1 Å) that likely functions as a gate. Constriction at this position is absent in other structurally characterized nAChRs but was observed in the bacterial Cys-loop receptor homolog ELIC in which the pore is constricted by a Phe

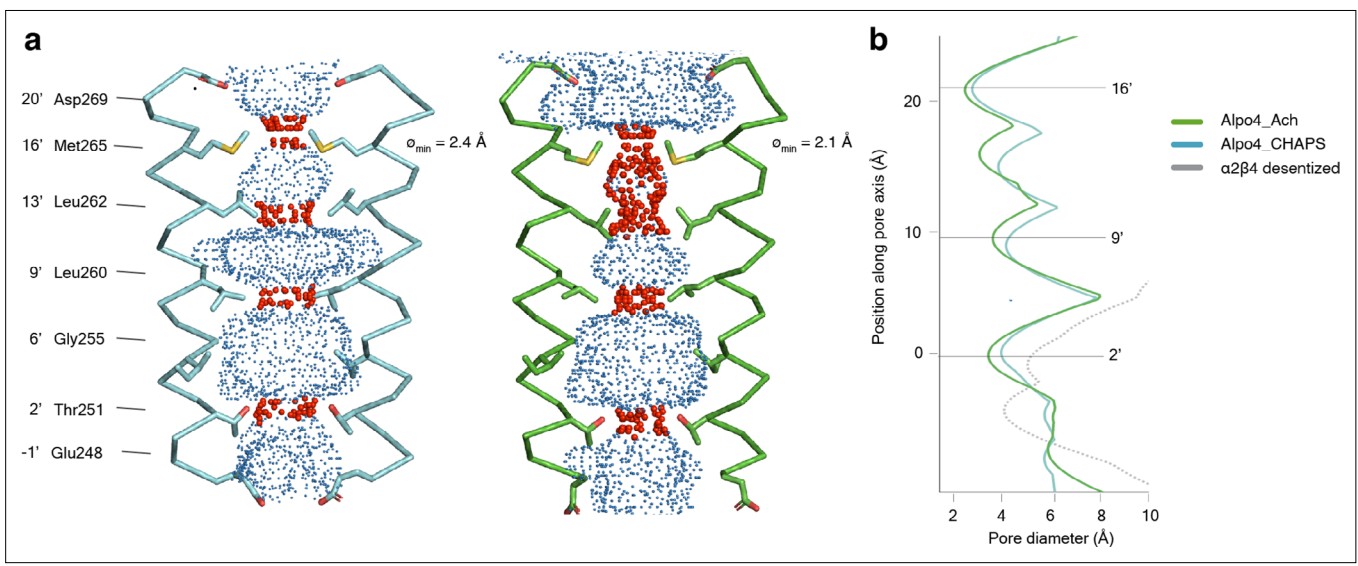

**Figure 4.** Permeation pathway of Alpo4. (**a**) Pore diameter calculated using HOLE and represented as dots for Alpo4[CHAPS] (blue) and Alpo4[APO] (green). Only M2 is shown in the cartoon and the pore-facing residues are shown as sticks. Constrictions are shown in red. (**b**) Pore diameter along the channel axis for Alpo4[CHAPS] (blue), Alpo4[APO] (green), and α4β2 (gray, PDB: 5KXI). The zero value along the channel axis corresponds to position 2' (Thr251).

The online version of this article includes the following source data for figure 4:

**Source data 1.** The pore diameter of the channels shown in panel b is calculated by HOLE.

residue at the extracellular surface (*Ulens et al., 2014*). This observation led us to speculate that the narrow 16' constriction prevented ion permeation, leading to a lack of agonist responses in our earlier electrophysiological ligand screenings. Therefore, we explored an M16'L mutation, which unfortunately was also unresponsive to acetylcholine or serotonin (data not shown). Next, the M16'L mutation was combined with the well-described L9'T mutation, which slows desensitization, converts certain antagonists into agonists, and increases $Ca^{2+}$ permeability in α7 nAChRs (*Galzi et al., 1992*). In our experiments, the double mutant M16'L/L9'T was still unresponsive to acetylcholine or serotonin. Additionally, we constructed chimeras in which the ECD and TMD were swapped with the α7 nAChR, similar to the α7/5-HT$_3$ chimera (*Eiselé et al., 1993*), but these constructs also remained unresponsive. Finally, we also considered that the 16' methionine residues could confer redox-sensitive channel regulation, similar to the upper gate formed in TRPV2 ion channels (*Fricke et al., 2019*). However, we could not detect any channel activity in the presence of oxidizing ($H_2O_2$) or reducing agents (DTT) (data not shown).

In conclusion, Alpo4 has a pore structure consistent with a cation-selective channel, but with an unusually tight constriction at the 16'M position, the role of which remains unclear.

## In silico ligand screening

To deorphanize Alpo4, we performed virtual screening using 37,000 compounds on three conformations of the Alpo4 receptor (see Methods for details). For compounds that were identified across multiple simulations, binding energies were calculated and ranked accordingly to select hit compounds with the greatest likelihood of producing agonistic activity. After examining the docking poses of the hits, only compounds docked within the ligand binding cavity were retained (*Figure 2—figure supplement 2* and *Supplementary file 2*).

Several of the top compounds contain sterol-like moieties including the top hit from virtual screening, ZINC36126889, which has an average binding energy of –12.1 kcal/mol (*Supplementary file 2*). This is a natural product synthesized by various species of Solanum plants and contains the sapogenin backbone structure. Metagenin also contains the sapogenin backbone, however, is more hydroxylated than ZINC36126889 and has an average binding energy of –11.3 kcal/mol. These compounds are structurally similar to diosgenin – a bioactive steroid sapogenin that is synthesized by a range of plant species (*Jesus et al., 2016*) and has been shown to bind in the orthosteric pocket of the chemo-tactile receptor from the striped pyjama squid (*Sepioloidea lineolata*) (*van Giesen et al., 2020*). As diosgenin is readily accessible, it has been extensively researched and acts at structurally related receptors where it was used as a substitute for ZINC36126889 and metagenin in functional studies. Bemcentinib, proscillaridin, and adapalene were also selected to examine functionally through electrophysiology and have predicted binding energies ≤ –11 kcal/mol. Based on their structural similarity to diosgenin, we further added the following analogs to our selection of compounds for testing in the electrophysiology assay: CHAPS, CHAPSO, and cholesteryl hemisuccinate (CHEMS). Finally, we also performed docking simulations with the neurotransmitters acetylcholine, GABA, and glycine, which were previously tested in a functional screen (*Wijckmans et al., 2016*).

To determine the agonist activity of hit compounds identified from virtual screening, all compounds were applied to oocytes injected with Alpo4 RNA. Mean currents elicited by a 10 s application were compared to uninjected cells, and compared to an application of a control solvent (ND96, EtOH, or DMSO). No compounds were found to elicit significantly different currents between Alpo4 or uninjected cells, with the exception of diosgenin. The highest concentration (100 µM) of diosgenin was found to elicit a significantly greater current in uninjected cells compared to Alpo4, which suggests it may be disrupting the cell membrane. This is in line with the structurally related compound glycodiosgenin, which has an estimated CMC of ~18 µM (*Chae et al., 2012*). This suggests none of the compounds tested are agonists of Alpo4 or, are unable to activate channel gating when Alpo4 is expressed in *Xenopus* oocytes.

## Discussion

Here, we described the structure of a lophotrochozoan homopentameric Cys-loop receptor, Alpo4. The structure of the nACh-like receptor from extreme thermophile *Alvinella pompejana* was solved in

apo form and in complex with CHAPS and provided insight into the architecture of the channel and its unexpected interaction with the sterol derivative.

## A possible function of Alpo4

Recent structural work on channels such as *Torpedo* nAChR, zebrafish GlyRα1, mouse 5-HT₃R, and human α7 nAChR revealed that representative Cys-loop receptors do not share a single conserved gating mechanism, highlighting their functional diversification (*Noviello et al., 2021*; *Polovinkin et al., 2018*; *Zarkadas et al., 2022*; *Kumar et al., 2020*). In lophotrochozoans, the vast expansion of nAChR genes, their high sequence diversity, and spatial expression profiles imply diversified function (*Jiao et al., 2019*). Our phylogenetic analysis revealed that Alpo4 doesn't cluster with homolog lophotrochozoans receptors, except for those found in *Capitella teleta*, suggesting a faster or less constrained evolution of the channel. Multiple sequence alignment shows the presence of key aromatic ligand-binding residues and the absence of the characteristic vicinal disulfide in the tip of loop C indicating that Alpo4 lacks signature amino acids of an α-like nAChR subunit (*Albuquerque et al., 2009*). The loss of the CC-tip is also observed in the Alpo3-like polychaetes receptors but not in the other lophotrochozoans organisms (*Figure 1—figure supplement 1b*).

Despite exhaustive screening efforts to identify the ligand, this Cys-loop receptor remains an orphan. Our efforts to identify its ligand included expression in *Xenopus* oocytes followed by two-electrode voltage-clamp electrophysiology with a library of >30 compounds known to act on various Cys-loop receptors (*Wijckmans et al., 2016*; *Galzi et al., 1992*). As a complementary approach, we expressed Alpo4 in HEK293 cells, followed by patch-clamp electrophysiology with a selected compound library, and employed lipid vesicles to reconstitute detergent-purified Alpo4. Virtual compound screening using determined structures identified potential binders of which the top hits had reminiscent flat moieties made of aromatic rings which were docked in the hydrophobic groove at the subunit interface. Some of these compounds were natural products. Nonetheless, none of them elicited a current response in two-electrode voltage-clamp experiments.

Neither electrophysiological recordings nor structural evidence indicated that either acetylcholine or serotonin binds Alpo4 or induces conformational changes in the protein.

Alpo4 might be gated by a ligand different from acetylcholine and serotonin consistent with the expansion of nACh genes in lophotrochozoan. Therefore, the agonist may be a chemical that is different from the currently known neurotransmitters in this family of ion channels (*Wijckmans et al., 2016*). It has recently been discovered that other marine organisms such as octopus and squid have nACh-like chemotactile receptors that do not respond to acetylcholine but are gated by poorly soluble terpenes and chloroquine (*van Giesen et al., 2020*). Like Alpo4, these receptors lack the common signature ligand motifs, supporting our proposal that Alpo4 is a functional channel for which the ligand remains unknown.

Furthermore, cryo-EM structure-guided mutagenesis of the pore residues and construction of chimeras did not result in an acetylcholine- or serotonin-responsive channel. Additionally, the 16'M pore constriction resembles the formation of an upper gate in TRPV2, which confers redox-sensitive channel regulation (*Fricke et al., 2019*). Oxidizing or reducing agents also failed to elicit a response from Alpo4. It is worth noting, that 16'M is not a thermophile-specific residue, as it is also present in the sequences of nAChRs-like proteins from mesophile organisms such as *Pecten maximus* and *Gigantopelta aegis*. It is possible that the channel adapted to extreme environmental conditions and remains closed under laboratory conditions. This, however, contradicts the electrophysiological experiments that allowed us to identify glycine as the agonist for Alpo5 and Alpo6, which displayed high sequence similarity to glycine receptors (45–50%) (*Wijckmans et al., 2016*), whereas Alpo7 was identified as a pH-gated ion channel (*Juneja et al., 2014*). This indicated that thermophilic Alpo Cys-loop receptors can be produced in the functional form in mesophilic expression systems highlighting Alpo4 as an outlier. Among other plausible reasons for the lack of functional activity of Alpo4 in electrophysiological experiments might be a requirement for an accessory β-subunit or a chaperone protein, similar to the recent identification of TMX3 as a co-factor required for the expression of insect nAChRs (*Ihara et al., 2020*).

Another unusual feature of the apo structure of Alpo4 is the presence of His131 on loop E, which forms a cation-π interaction with the highly conserved loop B aromatic W159 in the orthosteric binding site. The histidine could potentially coordinate a metal ion or require (de)protonation for the channel

to become responsive to an agonist. In the case of CHAPS, this interaction is broken enabling coordination of the quaternary ammonium group by W159. Although CHAPS is not expected to be a native ligand in *Alvinella pompejana,* our structures reveal it interacts with the ECD and induces a quaternary twist movement.

## Bivalent channel modulator

Pharmaceutical α7 nAChR agonists are composed of three representative groups: a cationic center, a hydrogen bond acceptor, and a hydrophobic element (*Mazurov et al., 2006*). These pharmacophores are often small molecules that fit in the orthosteric ligand-binding pocket. Here, the example of CHAPS binding creates an unusual precedent for the binding of a channel modulator that interacts with two regions of the ECD, one in the conserved ligand-binding pocket and another with the poorly conserved crevice outside the orthosteric site (*Figures 1, 3f and g*, *Figure 2—figure supplement 1b*).

When binding sites of larger known binders, including VHH (*Brams et al., 2020*; *Hénault et al., 2019*) and α-bungarotoxin (*Noviello et al., 2021*; *Rahman et al., 2020*) were examined (*Figure 3—figure supplement 1a*) a nanobody bound to ELIC in the site covering the sterol-binding groove was identified, however, its interactions with ELIC did not overlap significantly with the interior of the sterol-binding groove. This suggests that the latter is a novel target location for binders.

The sterol group connected by a linker binds in between subunits and induces conformational changes which also change the width of the groove in Alpo4 (*Figure 3f and g*), therefore, it likely plays an active role in the observed quaternary twist. The changes in the groove shape are not specific to Alpo4 but are also observed, for example, in the nicotinic α7 receptor (*Figure 3—figure supplement 1b*) suggesting that the groove can be targeted for allosteric modulation of the channel.

This finding hints at a possible strategy for designing specific Cys-loop channel modulators wherein the orthosteric binder is complemented by a chemical group binding at the interface between the subunits. In most channels, a crevice is present at the subunit interface (*Figure 2—figure supplement 1b*), permitting a design of a specific binder. Because the interface is poorly conserved between the nAChRs, the binding of the second group can be designed specifically for a particular channel and particular conformation thereby increasing the specificity of a pharmacologically active molecule.

While this manuscript was under revision, the structures of chemotactile receptors (CRs) CRT1 from octopus and CRB1 from squid, which do not respond to acetylcholine but instead to terpenes and bitter tastants like denatonium, were reported (*Kang et al., 2023*; *Allard et al., 2023*). Curiously, CRT1 was found to bind and be activated by steroid-like diosgenin moiety and similar molecules that structurally resemble CHAPS. Nonetheless, their mode of binding to CRT1 was different from that of CHAPS to Alpo4, and the sterol-binding groove was not involved in the interactions.

This study contributes to a better understanding of Cys-loop receptors in lophotrochozoans, highlighting Alpo4 as a member with a yet-to-be-identified neurotransmitter agonist. Our findings that sterol derivatives bind at the orthosteric binding site hint toward new strategies for designing specific channel modulators.

## Materials and methods

**Key resources table**

| Reagent type (species) or resource | Designation | Source or reference | Identifiers | Additional information |
|---|---|---|---|---|
| gene (*Alvinella pompejana*) | Alpo4 | GenBank | PLoS ONE 11(3): e0151183. doi:10.1371/journal.pone.0151183 | N/A |
| Cell line (*Spodoptera frugiperda*) | Sf9 | Thermo Fisher | CAS: 1149015 | Insect cells used for baculovirus production and expression of Alpo4 |

*Continued on next page*

*Continued*

| Reagent type (species) or resource | Designation | Source or reference | Identifiers | Additional information |
|---|---|---|---|---|
| chemical compound, drug | Lauryl Maltose Neopentyl Glycol; LMNG | Anatrace | CAS: 1257852-96-2 | Detergent used for the solubilization and purification of Alpo4 |
| chemical compound, drug | CHAPS (3-((3- cholamidopropyl) dimethylammonio)–1- propanesulfonate) | Anatrace | CAS: 75621-03-3 | Detergent used for the solubilization and purification of Alpo4 |
| chemical compound, drug | Acetylcholine chloride | Merck | CAS: 60-31-1 | Neurotransmitter |
| chemical compound, drug | Serotonin hydrochloride | Sigma-Aldrich | CAS: 153-98-0 | Neurotransmitter |
| chemical compound, drug | Graphene oxide | GO Graphene | N/A | Graphene Oxide Dispersion (1% Aqueous) |
| software, algorithm | CLC sequence manager 21.0.5 | Qiagen | N/A | |
| software, algorithm | SerialEM 3.0.8 | *Mastronarde, 2005* | RRID:SCR_017293 | |
| software, algorithm | MotionCorr2 | *Zheng et al., 2017* | RRID:SCR_016499 | |
| software, algorithm | CTFFIND-4.1 | *Rohou and Grigorieff, 2015* | RRID:SCR_016732 | |
| software, algorithm | crYOLO 1.7.0 | *Wagner et al., 2019* | RRID:SCR_018392 | |
| software, algorithm | Relion 3.0 | *Zivanov et al., 2018* | RRID:SCR_016274 | |
| software, algorithm | cryoSPARC 2.11, cryoSPARC 3.2.0 | *Punjani et al., 2017* | RRID:SCR_016501 | |
| software, algorithm | UCSF Chimera 1.13.1 | *Pettersen et al., 2004* | RRID:SCR_004097 | |
| software, algorithm | Coot 0.9 | *Casañal et al., 2020* | RRID:SCR_014222 | |
| software, algorithm | PHENIX 1.14 | *Liebschner et al., 2019* | RRID:SCR_014224 | |
| software, algorithm | MolProbity | *Williams et al., 2018* | RRID:SCR_014226 | |
| software, algorithm | ConSurf server | *Ashkenazy et al., 2016* | RRID:SCR_002320 | |
| software, algorithm | UCSF ChimeraX 1.3 | *Goddard et al., 2007* | RRID:SCR_015872 | |
| software, algorithm | The PyMOL Molecular Graphics System, Version 2.4.1 | Schrödinger, LLC | RRID:SCR_000305 | |
| Other | Quantifoil R2/1 Cu300 holey carbon grids | Quantifoil | N1-C15nCu30-01 | Electron microscopy grids used as a support to vitrify Alpo4 for cryo-EM studies |
| Cell line (*Xenopus laevis*) | *Xenopus laevis* oocytes | CRB Xénopes | XB-LAB-462 | Oocyte cells used for expression and functional characterization of Alpo4 |

*Continued on next page*

*Continued*

| Reagent type (species) or resource | Designation | Source or reference | Identifiers | Additional information |
|---|---|---|---|---|
| chemical compound, drug | Adapalene | Tokyo Chemical industry (TCI) | CAS: 106685-40-9 | |
| chemical compound, drug | Bemcentinib | TargetMol | CAS: 1037624-75-1 | |
| chemical compound, drug | Proscillaridin A | Sigma-Aldrich | CAS: 466-06-8 | |
| chemical compound, drug | Diosgenin | Fluorochem | CAS: 512-04-0 | |
| chemical compound, drug | CHAPSO (3-[(3-Cholamidopropyl) dimethylammonio]–2-Hydroxy-1-Propanesulfonate) | Anatrace | CAS: 82473-24-3 | |
| chemical compound, drug | CHEMS Cholesteryl hemisuccinate; | Anatrace | CAS: 102601-49-0 | |
| chemical compound, drug | Glycine | Sigma-Aldrich | CAS: 56-40-6 | Neurotransmitter |
| chemical compound, drug | GABA (γ-Aminobutyric acid) | Sigma-Aldrich | CAS: 56-12-2 | Neurotransmitter |
| software, algorithm | MTiOpenScreen | Ressource Parisienne en BioInformatique Structurale | Nucleic Acids Res 1;43(W1):W448-54. DOI:10.1093/nar/gkv306 | Online server used for virtual screening |
| software, algorithm | AutoDock Vina | Dr. Oleg Trott in the Molecular Graphics Lab (CCSB) at The Scripps Research Institute. | J. Chem. Inf. Model. 2021, 61, 8, 3891–3,898. DOI: 10.1021/acs.jcim.1c00203. | Software used for docking |
| Commercial assay, kit | mMESSAGE mMACHINE T7 ULTRA Transcription Kit | Invitrogen | Catalog No. AM1345 | Kit used to transcribe RNA for functional studies of Alpo4 |

## Phylogenetic tree construction

A sequence similarity search using BLASTP and the amino acid sequence of Alpo4 as a reference was performed against the genome of *Capitella teleta* (Polychaetes), the closest match to Alpo4, and other annelids with annotated genomes*: Dimorphilus gyrociliatus* (Polychaetes), *Owenia fusiformis* (Paleoannelids), *Hirudo verbana* (Clitellates), *Helobdella robusta* (Clitellates) and proteins of mollusca *Crassostrea virginca* (Bivalvia), *Crassostrea gigas* (Bivalvia), *Mizuhopecten yessoensis* (Bivalvia), *Pecten maximus* (Bivalvia), and *Pomacea canaliculata* (Gastropoda). Sequences with an E-value lower than 1e-5 were used for multiple sequence alignment using CLC v 21.0.5 sequence manager (Qiagen). After multiple rounds of alignments and manual removing non-nAChR sequences a set of 2047 proteins were obtained. To simplify the analysis, only one isoform of each receptor was retained, and a phylogenetic tree was constructed using the CLC sequence manager with the Maximum Likelihood method using 647 sequences. Confidence values were obtained using bootstrapping with 100 sequences. Here, family numbers were generated to identify which nAChRs cluster together as indicated on the phylogenetic tree. For the smaller phylogenetic trees of family 41 A (Alpo3-like) and family 43 A (Alpo4-like), only 1 isoform for each selected Lophotrochozoan member was used with the addition of *Octopus sensis* (OS), *Gigantopelta aegis* (GA), *Aplysia californica* (AC), and *Pomacea canaliculata* (PC).

## Protein expression and purification

Wild-type Alpo4 was expressed and purified as previously described with some modifications (*Wijckmans et al., 2016*). Briefly, His-tagged Alpo4 was expressed for 72 hr in Sf9 insect cells. Cells were pelleted, flash frozen, and stored at –80 °C. For protein purification, resuspended cells were lysed through high-pressure homogenization and membranes were isolated after ultracentrifugation.

Membranes were solubilized in 10 mM NaPi pH 7.4, 500 mM NaCl, 1% LMNG, and 1% CHAPS for 2 hr at 4 °C. The clarified supernatant was incubated with Ni-NTA resin (Roche), extensively washed with 10 mM NaPi pH 7.4, 500 mM NaCl, 0.05% LMNG, and 0.05% CHAPS and Alpo4 was eluted with the wash buffer containing 300 mM imidazole. The purity of the eluted fractions was assessed using SDS-PAGE and the fractions containing Alpo4 were concentrated in 100 kDa molecular weight cutoff concentrators (Invitrogen). Concentrated Alpo4 was injected into a Superose 6 column equilibrated with 25 mM NaPi, 150 mM NaCl, 0.007% CHAPS, and 0.003%, LMNG at 4 °C. For CHAPS-free Alpo4 preparations (Alpo4$^{APO\_LMNG}$, Alpo4$^{ACH}$, and Alpo4$^{SER}$), CHAPS was omitted from all the buffers used for the solubilization and purification.

## Cryo-EM sample preparation

Graphene oxide coated grids were prepared by incubating 4 µL of 0.9 mg/ml graphene oxide solution for 2 min on R2/1 grids freshly glow discharged for 1 min at 10 mbar pressure and 10 mA current. After blotting with Whatman 1 filter paper, the grids were washed three times with H$_2$0 and dried for 30 min. Grids with Alpo4$^{CHAPS}$ were prepared by applying 3 µL of Alpo4 (0.04 mg/mL) on the front and 1 µL on the back side of the grid and incubated for 1 min at 100% humidity and 25 °C. The sample was plunge-frozen using a CP3 (Gatan) after blotting for 2.7 s from both sides with Whatman 3 filter paper using a blotting force of –1. Grids that produced the Alpo4$^{APO}$ reconstruction were prepared using Alpo4 purified similar to Alpo4$^{CHAPS}$ sample, except that size-exclusion chromatography (SEC) was performed using a buffer without CHAPS. Next, 1 mM acetylcholine was added to the protein solution and incubated for 30 min on ice. A volume of 3 µL protein solution (0.04 mg/mL) was applied to the front of the graphene oxide-coated R2/1 grid and 1 µL of the buffer CHAPS-free SEC buffer was applied on the back of the grid. After 1 min incubation, the grid was blotted and plunge-frozen as described above. The grids with CHAPS-free Alpo4 samples (Alpo$^{APO\_LMNG}$, Alpo$^{ACH}$, and Alpo$^{SER}$) were prepared using the graphene oxide-coated grids by applying 2 µL of Alpo4 (0.05 mg/mL) solution on the front and 1 µL of the buffer on the back side of the grid and incubating for 1 min. Acetylcholine (1 mM) or serotonin (1 mM) were added to Alpo4$^{ACH}$, and Alpo4$^{SER}$ samples, respectively, and incubated for 30 min on ice prior to applying the protein solution on the grid. The grids were blotted as described above.

## Data collection

The data were collected on a JEOL CryoARM 300 transmission electron microscope (TEM) equipped with an Omega Filter (20 eV slit) using SerialEM v3.8.0 (*Mastronarde, 2005*). The Alpo4$^{CHAPS}$ dataset was collected on a Gatan summit K2 direct electron detector operating in counting mode. 5003 movies were collected using a nominal magnification of 60,000 (the calibrated pixel size of 0.782 Å/pixel) using a defocus range of 1.6–2.8 µm. Each movie consisted of 50 frames with 0.2 sec/frame exposure and was recorded with an electron flux of 3.7 e$^-$/s/Å$^2$. The Alpo4$^{APO}$ dataset comprised 7153 movies collected at a nominal magnification of 60,000 (the calibrated pixel size of 0.784 Å/pixel) on a K3 direct electron detector using a defocus range of 0.8–2.8 µm. Each movie contained 61 frames of 0.038 s exposure each using a dose of 18.2 e$^-$/s pixels. The Alpo4$^{APO\_LMNG}$, Alpo4$^{ACH}$, and Alpo4$^{SER}$ datasets comprised 11,895, 13,850, and 2201 movies, respectively collected at a nominal magnification of 60,000 (the calibrated pixel size of 0.760 Å/pixel) using a defocus range of 1.0–2.4 µm. Each movie contained 59 frames of 0.05 s exposure with a dose of 18.2 e$^-$/s pixels.

## Cryo-EM image processing

For all datasets, movie alignment was done with MotionCorr2 (*Zheng et al., 2017*) and contrast transfer function (CTF) was estimated with CtfFind4 (*Rohou and Grigorieff, 2015*). For the Alpo4$^{CHAPS}$ dataset, particles were picked with crYOLO v1.8 (*Wagner et al., 2019*), extracted in a box size of 368 pixels, and decimated four times (3.1 Å/pixel) followed by two rounds of 2D classification in Relion 3.0 (*Zivanov et al., 2018*) with between 20 and 50 classes. An *ab initio* model was generated without imposing symmetry and used as a starting model after low pass filtering to a resolution of 15 Å. Multiple rounds of 3D classification were performed with 3–6 classes without applying symmetry in Relion 3.0. The best-resolved classes were auto-refined in Relion imposing C5 symmetry. The selected particles were re-extracted without binning and subjected to Bayesian polishing and per-particle

defocus refinement followed by a final 3D auto-refinement step (*Zivanov et al., 2019*). During the post-processing step, a soft mask was applied. Local resolution was estimated using Relion 3.0.

In the case of the Alpo4$^{APO}$ dataset, images were denoised using the JANNI software (*Wagner, 2019*) prior to particle picking using crYOLO. Particles were extracted in a box size of 358 pixels and imported into CryoSPARC v3.3.1 (*Punjani et al., 2017*) and five rounds of reference-free 2D classification were performed. Five *ab initio* models were generated with C1 symmetry and a maximum resolution of 12 Å. Subsequent hetero- and non-uniform refinement with C5 symmetry and a dynamic mask resulted in a model with a resolution of 3.4 Å. Both global and local CTF refinement was performed in CryoSPARC. The particles were used for a final *ab initio* model calculation followed by non-uniform refinement with C5 symmetry. This resulted in a 3D reconstruction at a resolution of 3.4 Å. Similar processing strategies were applied to the Alpo$^{APO\_LMNG}$, Alpo$^{ACH}$, and Alpo$^{SER}$ datasets and resulted in reconstructions at a resolution between 4.1 and 6.1 Å (*Supplementary file 1*, *Figure 1—figure supplements 3 and 4*, *Figure 1—figure supplement 6* and *Figure 1—figure supplements 8–10*).

## Cryo-EM model building and structure analysis

The model of mouse serotonin 5-HT$_3$ receptor (PDB 6HIQ *Polovinkin et al., 2018*) was used as an initial model for building the structure of Alpo4$^{CHAPS}$. A monomer was fitted into the cryo-EM map (JiggleFit) and manually rebuilt using Coot 0.9.5 (*Emsley et al., 2010*). The model was expanded to a pentamer by applying the C5 symmetry operator in Phenix 1.19 (*Liebschner et al., 2019*). The resulting model was refined in Phenix using real_space_refinement routine (*Brown et al., 2015*). Here, global minimization, rigid body fit, and local rotamer fitting were performed with C5 symmetry imposed. After each refinement cycle, the model was manually adjusted in Coot. Due to the poor density of helix M4, it was initially built as a polyalanine chain. Later, the register of the M4 helix was determined from the 3.4 Å Alpo4$^{APO}$ reconstruction and fitted into Alpo4$^{CHAPS}$ map. The refined Alpo4$^{APO}$ model was fitted in Alpo$^{APO\_LMNG}$, Alpo$^{ACH}$, and Alpo$^{SER}$ reconstructions as a rigid body and refined using the real_space_refinement routine. The models were validated using MolProbity *Chen et al., 2010* , Phenix *Liebschner et al., 2019*, and Coot. Figures were generated using UCSF Chimera and UCSF ChimeraX v1.3 (*Goddard et al., 2007*), and PyMOL v2.4.0. Structure-based sequence alignment was performed in PROMALS3D (*Pei et al., 2008*).

## Virtual screening

Virtual in silico screening was performed using AutoDock Vina 2.4.6 (*Trott and Olson, 2010*) via the open-access MTi OpenScreen servers (*Labbé et al., 2015*). In total 37,137 compounds were screened across 97 simulations, utilizing 3 variations of the Alpo4 structure; (1) Apo structure (APO), (2) CHAPS-bound structure with CHAPS removed (CHAPS), (3) Apo structure with loop C extended 6.5 Å relative to S201 (loop C), which was included to enhance accessibility to the ligand-binding site during simulations. Six variations of the sampling grid were selected to ensure adequate sampling of the full ligand binding domain. The first two grids used the CHAPS ligand-binding pocket with a grid centered at x: 91.72, y: 125.29, z: 78.59, and dimensions of (1) 50 Å × 50 Å×50 Å or (2) 30 Å × 30 Å×30 Å. Grid location was calculated using the AutoDock plugin for PyMOL *Trott and Olson, 2010*. An additional 4 grids were designed by selecting residues that come within (3) 6 Å or (4) 4 Å of ACh within the ACh-binding protein (*Celie et al., 2004*), (5) residues that come within 4 Å of CHAPS in Alpo4, and (6) residues that come within 4 Å of classical neurotransmitters in pLGICs (ACh *Morales-Perez et al., 2016*, Glycine *Yu et al., 2021*, 5-HT *Basak et al., 2018*). All residue distances were calculated using PyMOL 2.5.2.

The compound libraries consisted of two in-built server libraries; (1) Diverse-lib, (2) Drug-lib, and five user-made libraries synthesized from PubChem's chemical database. User-made libraries were designed based on 3D structural similarity to (3) classical neurotransmitters, (4) CHAPS and sterol derivatives, and (5-7) agonists from the structurally related chemo-tactile receptors identified from Octopus bimaculoides (*van Giesen et al., 2020*; *Allard et al., 2023*) and Sepioloidea lineolata (*Kang et al., 2023*). Hit compounds that were identified across multiple simulations were collated, and those with the lowest binding energy that is readily available were purchased for electrophysiology studies. A sub-set of neurotransmitters and CHAPS-related compounds were also functionally tested.

All compounds selected for electrophysiology studies were re-docked using AutoDock Vina in USCF Chimera (*Trott and Olson, 2010*; *Labbé et al., 2015*; *Celie et al., 2004*; *Yu et al., 2021*;

*Eberhardt et al., 2021*) to validate hit compounds and to compare the binding of hit compounds to CHAPS and classical neurotransmitters. This process follows the methods outlined by Rabaan and colleagues (*Rabaan et al., 2023*). The CHAPS Alpo4 structure and all 12 compounds were prepared for docking using the in-built Dock Prep tool with default setting (*Shapovalov and Dunbrack, 2011*; *Wang et al., 2006*), which adds in polar hydrogens and charges. Re-docking of the 12 compounds was conducted using the AutoDock Vina chimera plug-in using a grid enclosing the CHAPS binding site, centered at the aforementioned position with dimensions of 30 Å × 27 Å × 30.5 Å. This identified the 5 top binding modes of each compound and calculated their predicted binding energies.

### Alpo4 expression in *Xenopus laevis* oocytes

All surgical procedures for *Xenopus laevis* oocyte extraction were conducted as previously described (*Nys et al., 2022*). Briefly, female *Xenopus* were anesthetized with 3-aminbenzoic acid ethyl ester for 12–15 min. Small incisions (1–1.5 cm) were made in the skin and muscle layers, in the lateral portion of the lower abdomen, and ovarian sacs were extracted using forceps. Ovarian sacs were mechanically disrupted, and oocytes were enzymatically dissociated from follicular cells using Type-I collagenase (Sigma). Oocytes were stored at 4 °C in MBS solution (0.7 mM $CaCl_2$, 5 mM hemi-$Na^+$-HEPES, 88 mM NaCl, 1 mM KCl, 2.5 mM $NaHCO_3$, 10 mM $MgSO_4$, pH 7.8) supplemented with 0.5 mM theophylline and 2.52 µM gentamicin.

The cDNA of Alpo4 was subcloned into pGEMHE and linearized using the *NheI* restriction enzyme (Thermo Fisher). mRNA was transcribed with the T7 RNA polymerase using the Invitrogen mMES-SAGE mMACHINE T7 ULTRA Transcription Kit (Thermo Fisher). Defolliculated stage IV oocytes were microinjected with 5.4 ng of mRNA and stored in supplemented MBS (see above) on an orbital shaker (PSU-10i, Grant-bio) at 14–16°C for 3–5 days.

### Two-electrode voltage clamp electrophysiology

Receptor activity was assessed using the automated two-electrode voltage clamp electrophysiology HiClamp system (Multichannel System, Germany). Whole-cell currents were measured using microelectrodes fabricated from thin-walled borosilicate capillary tubes (1 mm O.D, 0.75 mm I.D., World precision instruments) using a double-stage programmable horizontal pipette puller (PUL-1000, World precision instruments), and filled with 3 M KCl. Microelectrodes displayed typical resistance between 0.5 and 1.5 mOhms.

Electrophysiological recordings were conducted at 16 °C. Cells were held at –60 mV and continually perfused with ND96 (96 mM NaCl, 2 mM KCl, 1 mM $MgCl_2$, 1.8 mM $CaCl_2$, 5 mM hemi-$Na^+$-HEPES, pH 7.6). Data were acquired at 1000 Hz, filtered at 500 Hz, and analyzed using proprietary software running under Matlab (Mathworks Inc). Test solutions were dispersed in a 96-microtiter plate (NUNC, Thermo Fisher) and applied for 10 s with ≥2 min between subsequential applications. All test compounds were tested at ranges below their critical micelle concentration and maximal volumes of solvents used were additionally applied to injected and un-injected cells in the absence of compound. Water-soluble compounds (ACh, GABA, Gly, CHAPS, CHAPSO) were dissolved directly into ND96 whilst non-soluble compounds were dissolved in either DMSO (bemcentinib, adapalene) or EtOH (proscillaridin, diosgenin, cholesteryl hemi-succinate). Experiments were replicated n≥6 times, across ≥2 batches of cells obtained from different frogs. The activity of compounds on injected oocytes was compared to their activity on un-injected oocytes under the same conditions and was analyzed using a two-way ANOVA with Dunnett's multiple comparisons post-hoc test, with a significance value set to $p < 0.05$.

## Acknowledgements

We are indebted to Dr. Adam Schröfel and Dr. Marcus Fislage for their assistance with cryo-EM data collection at BECM. We thank Marijke Brams and Mieke Nys for assistance with baculovirus production and protein expression. We would like to acknowledge the funding provided by Vlaams Instituut voor Biotechnologie, Fonds Wetenschappelijk Onderzoek (Grant Nos. G0H5916N, G054617N to RGE, and G0H5916N to SDG). C U was supported by grants from FWO-Vlaanderen (G0C1319N, G087921N) and KU Leuven (C3/19/023).

## Additional information

### Funding

| Funder | Grant reference number | Author |
|---|---|---|
| Fonds Wetenschappelijk Onderzoek | G0H5916N | Steven De Gieter<br>Rouslan G Efremov |
| Fonds Wetenschappelijk Onderzoek | G054617N | Rouslan G Efremov |
| Fonds Wetenschappelijk Onderzoek | G0C1319N | Chris Ulens |
| Fonds Wetenschappelijk Onderzoek | G087921N | Chris Ulens |
| KU Leuven | C3/19/023 | Chris Ulens |

The funders had no role in study design, data collection and interpretation, or the decision to submit the work for publication.

### Author contributions

Steven De Gieter, Data curation, Formal analysis, Funding acquisition, Validation, Investigation, Visualization, Methodology, Writing - original draft, Writing – review and editing; Casey I Gallagher, Formal analysis, Investigation, Visualization, Methodology, Writing – review and editing; Eveline Wijckmans, Diletta Pasini, Formal analysis, Investigation, Methodology; Chris Ulens, Rouslan G Efremov, Conceptualization, Data curation, Formal analysis, Supervision, Funding acquisition, Visualization, Project administration, Writing – review and editing

### Author ORCIDs

Steven De Gieter ⓘ http://orcid.org/0000-0002-2776-3791
Chris Ulens ⓘ http://orcid.org/0000-0002-8202-5281
Rouslan G Efremov ⓘ http://orcid.org/0000-0001-7516-8658

### Decision letter and Author response

Decision letter https://doi.org/10.7554/eLife.86029.sa1
Author response https://doi.org/10.7554/eLife.86029.sa2

## Additional files

### Supplementary files

• Supplementary file 1. Channel parameter values for anionic pentameric ligand-gated ion channels. Ion channel PDB identifiers, channel conformation, ligands, and narrowest constriction residues are listed.

• Supplementary file 2. The top hits from virtual screening. Name of the compounds, their CID number, and average binding energies across the top five docking poses are shown. Binding energies for acetylcholine, GABA, and glycine are shown for comparison.

• MDAR checklist

### Data availability

The cryo-EM density maps and atomic models generated in this study have been deposited in the PDB and EMDB database under accession codes: 8BYI / EMDB-16326 (Alpo4CHAPS), 8BXF / EMDB-16317 (Alpo4APO), 8BX5 / EMDB-16308 (Alpo4LMNG_APO), 8BXB / EMDB-16314 (Alpo4ACH), 8BKE / EMDB-16316 (Alpo4COMB), 8BKD / EMDB-16315 (Alpo4SER).

The following datasets were generated:

| Author(s) | Year | Dataset title | Dataset URL | Database and Identifier |
|---|---|---|---|---|
| De Gieter S, Efremov RG, Ulens C | 2023 | Alvinella pompejana nicotinic acetylcholine receptor Alpo4 in complex with CHAPS | http://www.rcsb.org/structure/8BYI | RCSB Protein Data Bank, 8BYI |
| De Gieter S, Efremov RG, Ulens C | 2023 | Alvinella pompejana nicotinic acetylcholine receptor Alpo4 in complex with CHAPS | https://www.ebi.ac.uk/emdb/EMD-16326 | EMDB, EMD-16326 |
| De Gieter S, Efremov RG, Ulens C | 2023 | Alvinella pompejana nicotinic acetylcholine receptor Alpo4 in apo state (Alpo4_apo, dataset 1) | http://www.rcsb.org/structure/8BXF | RCSB Protein Data Bank, 8BXF |
| De Gieter S, Efremov RG, Ulens C | 2023 | Alvinella pompejana nicotinic acetylcholine receptor Alpo4 in apo state (Alpo4_apo, dataset 1) | https://www.ebi.ac.uk/emdb/EMD-16317 | EMDB, EMD-16317 |
| De Gieter S, Efremov RG, Ulens C | 2023 | Alvinella pompejana nicotinic acetylcholine receptor Alpo4 in apo state (dataset 1) | http://www.rcsb.org/structure/8BX5 | RCSB Protein Data Bank, 8BX5 |
| De Gieter S, Efremov RG, Ulens C | 2023 | Alvinella pompejana nicotinic acetylcholine receptor Alpo4 in apo state (dataset 1) | https://www.ebi.ac.uk/emdb/EMD-16308 | EMDB, EMD-16308 |
| De Gieter S, Efremov RG, Ulens C | 2023 | Alvinella pompejana nicotinic acetylcholine receptor Alpo in apo state (dataset 2) | http://www.rcsb.org/structure/8BXB | RCSB Protein Data Bank, 8BXB |
| De Gieter S, Efremov RG, Ulens C | 2023 | Alvinella pompejana nicotinic acetylcholine receptor Alpo in apo state (dataset 2) | https://www.ebi.ac.uk/emdb/EMD-16314 | EMDB, EMD-16314 |
| De Gieter S, Efremov RG, Ulens C | 2023 | Alvinella pompejana nicotinic acetylcholine receptor Alpo4 in apo state (Alpo4_comb dataset 3) | http://www.rcsb.org/structure/8BXE | RCSB Protein Data Bank, 8BXE |
| De Gieter S, Efremov RG, Ulens C | 2023 | Alvinella pompejana nicotinic acetylcholine receptor Alpo4 in apo state (Alpo4_comb dataset 3) | https://www.ebi.ac.uk/emdb/EMD-16316 | EMDB, EMD-16316 |
| De Gieter S, Efremov RG, Ulens C | 2023 | Alvinella pompejana nicotinic acetylcholine receptor Alpo4 in apo state (Alpo4_LMNG_Serotonin dataset 4) | http://www.rcsb.org/structure/8BXD | RCSB Protein Data Bank, 8BXD |
| De Gieter S, Efremov RG, Ulens C | 2023 | Alvinella pompejana nicotinic acetylcholine receptor Alpo4 in apo state (Alpo4_LMNG_Serotonin dataset 4) | https://www.ebi.ac.uk/emdb/EMD-16315 | EMDB, EMD-16315 |

The following previously published datasets were used:

| Author(s) | Year | Dataset title | Dataset URL | Database and Identifier |
|---|---|---|---|---|
| Basak S, Chakrapani S | 2019 | Cryo-EM structure of 5HT3A receptor in presence of granisetron | http://www.rcsb.org/structure/6NP0 | RCSB Protein Data Bank, 6NP0 |

*Continued on next page*

*Continued*

| Author(s) | Year | Dataset title | Dataset URL | Database and Identifier |
|---|---|---|---|---|
| Morales-Perez CL, Noviello CM, Hibbs RE | 2016 | X-ray structure of the human Alpha4Beta2 nicotinic receptor | http://www.rcsb.org/structure/5KXI | RCSB Protein Data Bank, 5KXI |
| Sauguet L, Corringer PJ, Delarue M | 2013 | The GLIC pentameric Ligand-Gated Ion Channel at 2.4 A resolution | http://www.rcsb.org/structure/4HFI | RCSB Protein Data Bank, 4HFI |
| Spurny R, Govaerts C, Evans GL, Pardon E, Steyaert J, Ulens C | 2019 | X-ray structure of a pentameric ligand gated ion channel from Erwinia chrysanthemi (ELIC) 7'C pore mutant (L238C) in complex with nanobody 72 | http://www.rcsb.org/structure/6HJX | RCSB Protein Data Bank, 6HJX |

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
