## [Editor Report]

The authors solved cryoEM structural maps for the pLGIC homolog Alpo4 from an extreme thermophile worm in apo and CHAPS bound conditions. The data are convincing and valuable and reveal how a detergent can bind to the orthosteric site and induce a quaternary twist of the channel domains. A limitation is that it is difficult to relate these observations to channel function as the activating ligand for Alpo4 remains unknown.

---

## [Decision Letter]

**Decision letter after peer review:**

Thank you for submitting your article "Sterol derivative binding to the orthosteric site causes conformational changes in an invertebrate Cys-loop receptor" for consideration by *eLife*. Your article has been reviewed by 3 peer reviewers, including Marcel P Goldschen-Ohm as the Reviewing Editor and Reviewer #1, and the evaluation has been overseen by Richard Aldrich as the Senior Editor. The following individuals involved in the review of your submission have agreed to reveal their identity: Hugues Nury (Reviewer #2); Pierre-Jean Corringer (Reviewer #3).

Essential revisions:

1) The authors should consider the suggestion of Reviewer #2 to test for activation by CHAPS or a related steroid moiety in electrophysiology experiments.

2) As Reviewer #3 asked, were the structures used to aid in any computational searches for orthosteric ligands?

*Reviewer #2 (Recommendations for the authors):*

My bigger suggestion would be to try to assess the effect of CHAPS (or related compounds with a steroid moiety if CHAPS was too tricky because of its high CMC) in electrophysiology experiments. If I'm not mistaken, the authors do not report trying that. I know it might just dissolve the cell membrane -or there could be many other reasons it could fail- but the possible benefit for the manuscript is worth the effort. In a forthcoming publication also reviewed by this reviewer (from another group and on a different receptor) the authors tested the functional effect of a detergent molecule and found it acted as an agonist. A similar experiment might be appropriate here.

The main text figures are clear with a unified color code. Yet, in several instances, I caught myself thinking figures would benefit from additions/changes. The list below mixes minor points and maybe more important ones.

Would there be a way to show all the structures solved? Like a small panel in figure 1 showing there are X more structures in the apo conformation? This would help to visually relate the supplementary structures (AlpoSER, AlpoAPOLMNG) to the ones that are already depicted in the main figures. It took me a bit of time, in the beginning, to relate the PBD list to the main figures because there were 5 items in the list and mostly 2 in the figures.

Maybe different colors for the different structures of Figure 2 would help an immediate identification of the different receptors or conformations (e.g a+d in blues, c+f in greens, b+e in something else).

In Figure 2 or 3, would an electrostatic surface representation show a hydrophobic groove where the tail of CHAPS bind? (e.g. in a view similar to SI4b). Also, it would be interesting to see what happens to this 'cavity' in a surface representation of the apo state. Does it disappear, does it shape change?

Curiosity question. Why so many particles in the final ALPOAch set? 250k is >10-fold more than ALPOApo and I would have imagined that sub-classifications would improve the homogeneity and therefore the resolution of the reconstruction.

Maybe a relevant citation for the discussion on phylogeny would be van Giesen et al.

That paper looks at divergent pLGICs that perform chemosensation in octopus.

1. van Giesen, L., Kilian, P. B., Allard, C. A. H., and Bellono, N. W. (2020) Molecular Basis of Chemotactile Sensation in Octopus. Cell 183, 594-604.e14

*Reviewer #3 (Recommendations for the authors):*

The resolution of the fine architecture of the orthosteric site, especially with a bound CHAPS molecule, could help the selection of potential ligands by visual inspection and/or computational methods. This idea is however not developed in the paper, and it would be nice to hear from the authors if they explored this aspect, in particular in relation to their previous functional screening published in Plos One 2016.

---

## [Author Response]

Essential revisions:1) The authors should consider the suggestion of Reviewer #2 to test for activation by CHAPS or a related steroid moiety in electrophysiology experiments.

We have tested CHAPS and related sterol derivatives using two-electrode voltage clamp recordings on *Xenopus oocytes* expressing Alpo4. We could not observe robust or reproducible agonist effects of CHAPS and related compounds. This suggests that CHAPS is not an agonist for the channel. An additional paragraph describing virtual screening and mentioning the results of electrophysiological measurements has been added on page 12 line 245 and related Method’s section has been added on page 24.

2) As Reviewer #3 asked, were the structures used to aid in any computational searches for orthosteric ligands?

We have now conducted a virtual screening to identify potential ligands for Alpo4, but this screening did not reveal any compounds with agonist activity. A description of the screening and results of conductivity measurements has been added on page 12 from 254.

Reviewer #2 (Recommendations for the authors):My bigger suggestion would be to try to assess the effect of CHAPS (or related compounds with a steroid moiety if CHAPS was too tricky because of its high CMC) in electrophysiology experiments. If I'm not mistaken, the authors do not report trying that. I know it might just dissolve the cell membrane -or there could be many other reasons it could fail- but the possible benefit for the manuscript is worth the effort. In a forthcoming publication also reviewed by this reviewer (from another group and on a different receptor) the authors tested the functional effect of a detergent molecule and found it acted as an agonist. A similar experiment might be appropriate here.

We have performed virtual screening which identified many compounds, most of them comprised out of conjugated rings similar to CHAPS. The top hits, including CHAPS and its derivatives, were tested using two-electrode voltage clamp recordings on *Xenopus oocytes* expressing Alpo4. We could not observe robust or reproducible agonist effects of CHAPS and related compounds.

This suggests that CHAPS is not an agonist for the channel, which is consistent with our structural observations. The details of the experiments and a list of the tested compounds are added to the manuscript on p 12 from line 254, Figure 2—figure supplement 2 and Supplementary Table 2. Discussion p15 line 320 and Methods p 24 from line 526.

The main text figures are clear with a unified color code. Yet, in several instances, I caught myself thinking figures would benefit from additions/changes. The list below mixes minor points and maybe more important ones.Would there be a way to show all the structures solved? Like a small panel in figure 1 showing there are X more structures in the apo conformation? This would help to visually relate the supplementary structures (AlpoSER, AlpoAPOLMNG) to the ones that are already depicted in the main figures. It took me a bit of time, in the beginning, to relate the PBD list to the main figures because there were 5 items in the list and mostly 2 in the figures.

Figure 1 has been updated, now it shows maps and 3D models of all 5 obtained reconstructions including Alpo4^ACh^, Alpo4^LMNG^ and Alpo4^Ser^.

Maybe different colors for the different structures of Figure 2 would help an immediate identification of the different receptors or conformations (e.g a+d in blues, c+f in greens, b+e in something else).

As suggested by the reviewer, we have changed the colors for different conformations and receptors.

In Figure 2 or 3, would an electrostatic surface representation show a hydrophobic groove where the tail of CHAPS bind? (e.g. in a view similar to SI4b). Also, it would be interesting to see what happens to this 'cavity' in a surface representation of the apo state. Does it disappear, does it shape change?

We have added to Figure 3 panels f and g displaying electrostatic potential around CHAPS binding site for bound and apo states. Both show that the cavity remains hydrophobic, but its width is different between the 2 conformations.

Curiosity question. Why so many particles in the final ALPOAch set? 250k is >10-fold more than ALPOApo and I would have imagined that sub-classifications would improve the homogeneity and therefore the resolution of the reconstruction.

In the case of the Alpo4_ACh dataset, we were able to collect more movies and extract more particles than was the case of the Alpo4_Apo dataset. The average contrast was lower, however. That is why we think subsequent 3D classification steps did not result in subsets of particles producing higher-resolution reconstructions.

Maybe a relevant citation for the discussion on phylogeny would be van Giesen et al.That paper looks at divergent pLGICs that perform chemosensation in octopus.

This is an excellent suggestion. The paper “Molecular Basis of Chemotactile Sensation in Octopus” by van Giesen et al., 2020 reports the discovery of a cephalopod-specific chemotactile receptors (CRs). These pLGICs do not respond to acetylcholine and similar to Alpo4, lack the signature ligand binding CC-tip motif in the C-loop. The authors were able to identify specific molecules such as chloroquine and nootkatone that do elicit a channel response.

In the revised manuscript we mention this paper in the Discussion section p 16 line 332:

“It has recently been discovered that other marine organisms such as octopus and squid have nACh-like chemotactile receptors that do not respond to acetylcholine but are gated by poorly soluble terpenes and chloroquine^40^. Like Alpo4, these receptors lack the common signature ligand motifs, supporting our proposal that Alpo4 is a functional channel for which ligand remains unknown.”

We have also mentioned the manuscripts describing the structures of chemotictile receptors that were published while we were working on the revision. Page 18 from line 394:

“While this manuscript was under revision, the structures of chemotactile receptors (CRs) CRT1 from octopus and CRB1 from squid, which do not respond to acetylcholine but instead to terpenes and bitter tastants like denatonium, were reported ^50,51^. Curiously, CRT1 was found to bind and be activated by steroid-like diosgenin moiety and similar molecules that structurally resemble CHAPS. Nonetheless, their mode of binding to CRT1 was different from that of CHAPS to Alpo4 and the sterol-binding groove was not involved in the interactions.”

Reviewer #3 (Recommendations for the authors):The resolution of the fine architecture of the orthosteric site, especially with a bound CHAPS molecule, could help the selection of potential ligands by visual inspection and/or computational methods. This idea is however not developed in the paper, and it would be nice to hear from the authors if they explored this aspect, in particular in relation to their previous functional screening published in Plos One 2016.

We have now conducted virtual screening using apo Alpo4 and CHAPS-bound structures. The top hits identified in virtual screening are shown in Figure 2—figure supplement 2 and Supplementary Table 2. The top hits were tested using two-electrode voltage clamp recordings on *Xenopus oocytes* expressing Alpo4. We could not observe robust or reproducible agonist effects of CHAPS and related compounds. The details of the experiments are added to the manuscript on p 12 from line 254, p15 line 320 and Methods p 24.